# Elucidating the mechanism by which synthetic helper peptides sensitize *Pseudomonas aeruginosa* to multiple antibiotics

Yushan Xia[1,2], Rubén Cebrián[1], Congjuan Xu[2], Anne de Jong[1], Weihui Wu[2]*, Oscar P. Kuipers[1]*

**1** Department of Molecular Genetics, Groningen Biomolecular Sciences and Biotechnology Institute, University of Groningen, Groningen, Netherlands, **2** State Key Laboratory of Medicinal Chemical Biology, Key Laboratory of Molecular Microbiology and Technology of the Ministry of Education, Department of Microbiology, College of Life Sciences, Nankai University, Tianjin, China

* wuweihui@nankai.edu.cn (WW); o.p.kuipers@rug.nl (OPK)

**Data Availability Statement:** The transcriptome (RNA sequencing) data that support the findings of this study have been deposited in the NCBI

## Abstract

The emergence and rapid spread of multi-drug resistant (MDR) bacteria pose a serious threat to the global healthcare. There is an urgent need for new antibacterial substances or new treatment strategies to deal with the infections by MDR bacterial pathogens, especially the Gram-negative pathogens. In this study, we show that a number of synthetic cationic peptides display strong synergistic antimicrobial effects with multiple antibiotics against the Gram-negative pathogen *Pseudomonas aeruginosa*. We found that an all-D amino acid containing peptide called D-11 increases membrane permeability by attaching to LPS and membrane phospholipids, thereby facilitating the uptake of antibiotics. Subsequently, the peptide can dissipate the proton motive force (PMF) (reducing ATP production and inhibiting the activity of efflux pumps), impairs the respiration chain, promotes the production of reactive oxygen species (ROS) in bacterial cells and induces intracellular antibiotics accumulation, ultimately resulting in cell death. By using a *P. aeruginosa* abscess infection model, we demonstrate enhanced therapeutic efficacies of the combination of D-11 with various antibiotics. In addition, we found that the combination of D-11 and azithromycin enhanced the inhibition of biofilm formation and the elimination of established biofilms. Our study provides a realistic treatment option for combining close-to-nature synthetic peptide adjuvants with existing antibiotics to combat infections caused by *P. aeruginosa*.

## Author summary

Antimicrobial resistance is a global public health problem that is limiting our antibiotic capacity to control bacterial infections. Despite the enormous efforts in drug development, very few clinically relevant drugs have been approved in the last decades. This is especially critical for Gram-negative bacteria, which remain the most problematic group, because, among other reasons, the outer membrane acts as a permeability barrier preventing antibiotics to reach their targets. Here, we describe the action mechanism through

Sequence Read Archive (SRA) with the accession code PRJNA695137.

**Funding:** Y. X. was supported by the program of China Scholarships Council (No.201906200035) R. C. was supported by NACTAR program of Dutch NWO-TTW (Project No. 16433) The funders had no role in the study design, data collection and interpretation, or the decision to submit the work for publication.

**Competing interests:** The authors have declared that no competing interests exist.

which a human close-to-nature synthetic "helper" peptide is able to sensitize *Pseudomonas aeruginosa* to a broad range of antibiotics for which it is normally resistant. We demonstrate that this peptide is active at different levels (membrane, proton motive force, respiration, ATP synthesis, ROS accumulation and efflux pump activity), facilitating antibiotics accumulation inside the bacteria and promoting the cell death. Moreover, we have determined the efficacy of the synergistic combinations in complex relevant environments, such as *Pseudomonas* biofilms, human blood and *in vivo* in a mice infection model. This work demonstrates the effectiveness of synergistic combinations for fighting an important lung pathogen, providing a new therapeutic option for the treatment of *P. aeruginosa* infections and extending the use of the current antibiotics.

## Introduction

The development of multi-drug resistance is a global phenomenon that severely impairs the effectiveness of antibacterial chemotherapy. During the past decades, only a few new narrow-spectrum drugs have been developed, while the existing drugs are rapidly losing their efficacy [1,2]. These new drugs are generally based on chemical modifications of previous ones, so the development of resistance is just a matter of time since the resistance mechanisms are already established in nature [3,4]. Gram-negative bacteria remain the most challenging group since the outer membrane acts as a permeability barrier, which makes difficult for antibiotics to reach their intracellular targets [5]. For this reason, it is not surprising that *P. aeruginosa* has been pointed out as one of the most problematic Gram-negative pathogens for which new treatments are urgently needed. *P. aeruginosa* is a ubiquitous opportunistic pathogen that can cause a broad range of infections in humans, especially in ICU, immunodeficient, burn wound and cystic fibrosis patients (CF) [6]. *P. aeruginosa* has been found to be naturally resistant to several types of antibiotics because of its low membrane permeability, the production of antibiotic-modifying enzymes and the constitutive expression of multidrug efflux systems [7]. Besides, the formation of biofilms further impedes clinical treatments, especially in CF patients. The extracellular matrix of biofilms shields the bacteria from antibiotics, making the embedded bacteria up to 1,000 times more resistant [8].

In the era of antimicrobial resistance, the combined use of drugs to enhance the antimicrobial activity provides an alternative therapeutic option to extend the use of narrow-spectrum antibiotics of Gram-positive bacteria to Gram-negative bacteria or to increase the effectiveness of traditional Gram-negative bacteria-targeting antibiotics [9]. The boosted antimicrobial activity provided by the synergistic relation not only allows a fast clearance of the infection but also shortens the courses of the treatments, reduces side effects and toxicity and delays the evolutionary selection of resistant strains [10]. In this sense, the combination of peptides with different established antibiotics represents one of the most promising approaches to fight multidrug resistant bacteria. These molecules are typically short, cationic and amphipathic that selectively target microbial cells with a fast action mechanism and low frequency in selecting resistant strains [11] and that can be produced (or synthetically designed) by species of any domain of life. In fact, previous studies have shown that the human-related antimicrobial peptides as cathelicidin LL-37 can increase the bactericidal effects of azithromycin against MDR Gram-negative bacteria by increasing the permeability of the outer membrane [12]. The short peptide KR-12 analogue designed from LL-37 showed good antimicrobial activities without mammalian cell toxicity [13]. Besides, the peptide L-11 (derived from KR-12) showed a good synergistic effect with lipid II binding antimicrobials as nisin or vancomycin against Gram-

negative pathogens [14]. Other short antimicrobial peptides with different origins such as the synthetic peptides EC5 or RW-BP100 (which are rich in arginine and lysine residues), the peptide Bac8c, a variant derived from the naturally occurring bovine peptide bactenecin, or the insect-related antimicrobial peptides as Api1b, are known to display bactericidal effect against Gram-negative bacteria by binding the negatively charged outer membrane [15–18]. Due to the increase in resistance to traditional antibiotics, the development of antimicrobial peptides has gradually become a focus of research due to their antibacterial ability or synergy with antibiotics.

Here, we explore the combined effect of putative outer-membrane perturbing peptides which exert low to modest activity against Gram-negative bacteria and known low human cell toxicity, to enhance the antimicrobial activity of multiple antibiotics against the Gram-negative pathogen *P. aeruginosa*. The action mechanisms of both, the best-tested peptide and the combinations thereof with classical antibiotics were fully characterized, concerning the activities of such combinations on both planktonic cells and biofilms of *P. aeruginosa*. Moreover, their efficient antimicrobial action was confirmed by an *ex vivo* bacteremia model and an *in vivo* murine abscess infection model.

## Results

### Combination of synthetic peptides with azithromycin against *P. aeruginosa*

Macrolides are used generally for Gram-positive bacteria treatments since Gram-negative bacteria as *P. aeruginosa* are inherently resistant to them due to the low permeability of the outer membrane and the presence of the efflux pumps MexXY-OprM and MexAB-OprM [7]. Although *P. aeruginosa* exhibits strong tolerance to azithromycin *in vitro*, it is still the clinically recommended antibiotic for the treatment of *P. aeruginosa* infections *in vivo* especially in CF patients [19,20], since, at sub-MIC concentrations, azithromycin retards biofilm formation of *P. aeruginosa* [21] and it has shown anti-inflammatory and anti-virulence properties [22]. To improve the efficacy of azithromycin against *P. aeruginosa*, we tested the synergistic effects of 12 previously reported peptides with a wide range of individual activity against Gram-negative pathogens and known low human cell toxicity as well as several newly designed variants (with increased stability, amphiphilicity and hydrophobicity) with azithromycin against the wild type reference strain PAO1 (Table 1). Interestingly, peptides with good intrinsic antimicrobial activity such as Bac8c or RW-BP100 and their variants did not exert any synergistic effect with azithromycin, while non-effective peptides (MIC ≥32 μM) such as EC5, KR-12-a2, L-11 and some of their variants, demonstrated strong synergistic effects (Table 1). It is worth noting that five of the six peptides (except EC5) that exhibited synergistic effects were close-to-nature derivatives from the cathelicidin LL-37. KR-12-a2 is an analogue designed from KR-12 (residues 18–29 of LL-37) [13] and L-11 is a recently KR-12-a2 described derivative that enhances the activity of lipid II-active antibiotics against Gram-negative bacteria [14]. Due to the relative instability of the peptides with L-amino acids in plasma, some peptides were synthesized with D-amino acids to increase their resistance to proteases and peptidases [23]. When peptide EC5 was synthesized with D-amino acids, it lost the synergistic effect with azithromycin, suggesting a chiral-related activity for the combination in this case, while others such as D-11 and D-RW-BP100 peptides retained full activity (Table 1). The activity of these peptides was not even compromised by the reverse sequence of the N-terminal to C-terminal amino acids (D-11R and D-RW-BP100R) suggesting no stereo-specificity for the target. Finally, we explored the role of arginine/lysine and amphiphilicity/hydrophobicity for the most promising peptide, i.e., L-11 [24]. When the fifth arginine of L-11 was replaced by tryptophan (increasing amphiphilicity and hydrophobicity), the antibacterial activity of the resulting peptide (L-11-R5W)

**Table 1. Sequences and MICs (μM) for the peptides and azithromycin used in this work.**

| Peptides | MICa/MICac (μM) | MICb/MICbc (μM) | FICI | Sequence | Source |
|---|---|---|---|---|---|
| Api1b | 128/128 | >32/8 | 1.125 | GNNRPVYIPQPRPPHPRL | [18] |
| Oncocin | 128/128 | >32/8 | 1.125 | VDKPPYLPRPRPPRRIYNR | [25] |
| Pyrrhocoricin | 128/128 | >32/8 | 1.125 | VDKGSYLPRPTPPRPIYNRN-NH2 | [26] |
| EC5 | 128/8 | >32/8 | **0.188** | RLLFRKIRRLKR | [15] |
| D-EC5 | 128/128 | >32/8 | 1.125 | rllfrkirrlkr | This study |
| Bac8c | 128/128 | 4/1 | 1.250 | RIWVIWRR-NH2 | [16] |
| RW-BP100 | 128/128 | 8/2 | 1.250 | RRLFRRILRWL-NH2 | [17] |
| D-RW-BP100 | 128/128 | 8/2 | 1.250 | rrlfrrilrwl-NH2 | This study |
| D-RW-BP100R | 128/128 | 4/1 | 1.250 | lwrlirrflrr-NH2 | This study |
| LL-37 | 128/64 | 8/2 | 0.75 | LLGDFFRKSKEKIGKEFK**RIVQRIKDFLR**NLVPRTES | [27] |
| KR-12-a2 | 128/8 | 32/4 | **0.188** | KRIVQRIKKWLR-NH2 | [13] |
| L-11 | 128/4 | 32/4 | **0.156** | RIVQRIKKWLR-NH2 | [14] |
| D-11 | 128/4 | 32/4 | **0.156** | rivqrikkwlr-NH2 | [14] |
| D-11R | 128/4 | 32/4 | **0.156** | rlwkkirqvirk-NH2 | [14] |
| L-11-R5W | 128/128 | 4/1 | 1.250 | RIVQWIKKWLR-NH2 | This study |
| D-11-k(r) | 128/4 | 32/4 | **0.156** | rivqrirrwlr-NH2 | This study |
| PEP9 | 128/128 | >32/8 | 1.125 | NGVQPKYK | [28] |
| PEP10 | 128/128 | >32/8 | 1.125 | KIAKVALKALK | [29] |
| ADP-1 | 128/128 | 16/4 | 1.125 | GIGKHVGKALKGLKGLLKGLGEC | [30] |
| ADP-1a | 128/128 | >32/8 | 1.125 | LKGLKGLLKGLGEC | This study |
| ADP-1b | 128/128 | >32/8 | 1.125 | KHVGKALKGLK | This study |
| ADP-1c | 128/128 | >32/8 | 1.125 | LKGLKGLLKGL | This study |

MICa: the MIC of azithromycin, MICb: the MIC of the peptide, MICac: the MIC of azithromycin in the combination, MICbc: the MIC of the peptide in the combination, the synergy effects were bolded in FICI. Peptides with capital letters mean they are synthesized with L-type amino acids, peptides with lowercase letters mean they are synthesized with D-type amino acids, red marks mean amino acid changes. D-11 and its variants were derived from the sequence that was bolded in LL-37 peptide.

was enhanced 8-fold. However, the synergistic activity with azithromycin was lost. When replacing the two lysines of D-11 by arginine (D-11-k(r)) no effect was observed. Notably, in all these active combinations, the MIC for azithromycin was reduced from 128 μM to 4 μM (8 μM in the case of EC5 and KR-12-a2) (Table 1).

## The antimicrobial synergism is extendable to other macrolides

To explore whether the observed effect is restricted to azithromycin or extendable to other macrolide-related antibiotics, we tested the synergistic effect of the selected helper peptides and other macrolide-related antibiotics to extend their use against *P. aeruginosa*. The helper peptides sensitized the bacteria to erythromycin, clarithromycin and telithromycin, but not to spiramycin (S1 Fig and S1 Table). In fact, only D-11 and KR-12-a2 showed an additive effect for the combination with spiramycin. Based on these data and according to previous results [14] D-11 was selected for further characterization of its synergistic activity and action mechanism.

Standard checkerboard microdilution assays confirmed that 2 μM D-11 was sufficient to reduce the MIC of azithromycin by 16 times against PAO1 (Fig 1A). Most antimicrobial peptides exert a bactericidal effect within 1–2 hours and then lose their activities, also reducing the bacteria viability at sub-MIC concentrations. To verify the synergistic effect, we tested the killing kinetic for the peptide D-11 at 0, 2, 4, 8, 16, 32 μM in 6 h using the MHB medium. The

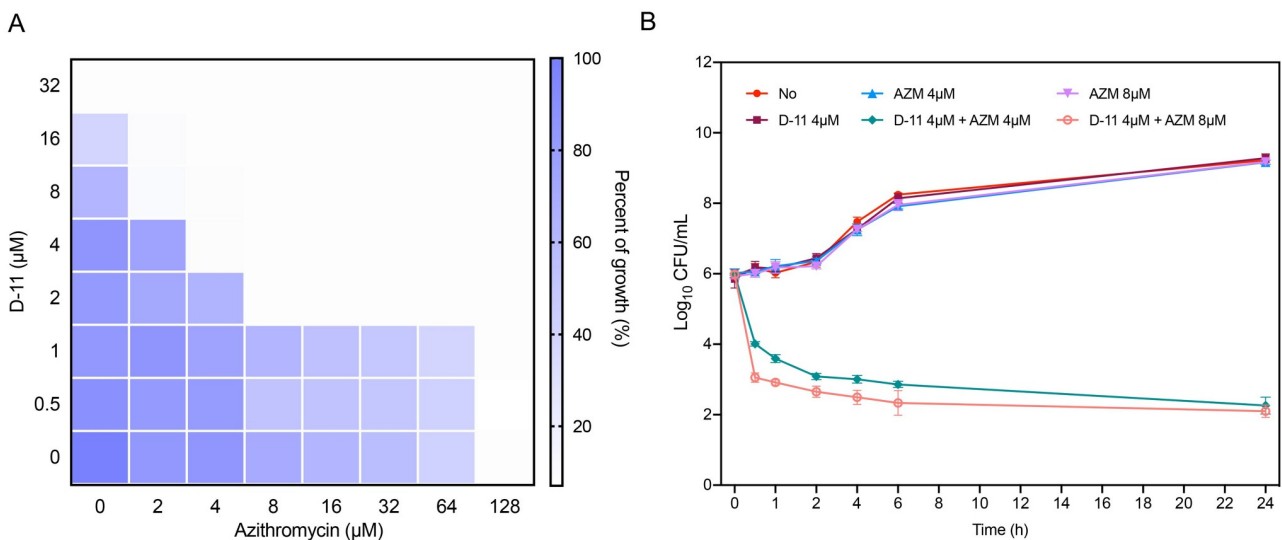

**Fig 1. D-11 synergizes with azithromycin against *P. aeruginosa*.** (**A**) Representative heat plots of the microdilution checkerboard assay for the combination of azithromycin and D-11 against PAO1. (**B**) Time killing curves for D-11 and azithromycin monotherapy and combination therapy against PAO1 during 24 h incubation at the indicated concentrations. Error bars represent mean ± SD for three biological replicates.

results showed that, at sub-MIC concentrations, D-11 could not kill the bacteria while at the MIC (32 μM) D-11 reducedPMF disruption measurement. CCCP was 99% bacterial viability (S2A Fig). When a sub-MIC concentration of D-11 (4 μM) was combined with AZM, the bacterial survival rate was significantly reduced (more than 99%), while the mono-treatment did not kill the bacteria (Fig 1B).

## D-11 induces outer- and inner membrane permeabilization by LPS and inner membrane phospholipids interaction

Previously we demonstrated that D-11 can increase the permeability of the outer membrane of a variety of Gram-negative bacteria [14], which however does not fully explain the observed synergism since efflux pumps are the most important mechanism related to macrolide resistance in *P. aeruginosa*. Therefore, we first measured the outer membrane permeability of PAO1 under different concentrations of D-11 treatment using the hydrophobic fluorescent probe 1-N-phenylnaphthylamine (NPN), with polymyxin B as a positive control. As expected, the outer membrane permeability of PAO1 increased in a dose-dependent manner (Fig 2A). It has been reported that some peptides not only increase the permeability of the outer membrane but also enhance the permeability of the inner membrane and cause changes in the membrane potential [31,32]. Therefore, we used the DNA-binding dye propidium iodide (PI) to detect the permeability of the inner membrane. The results showed that PAO1 exhibited dose-dependent uptake of PI by D-11, which indicates an increase in the permeability of the bacterial inner membrane (Fig 2B). Besides, we checked in the fluorescent microscope the membrane permeabilization of the bacteria to PI under the same conditions. As expected, PI-stained bacteria were observed in a dose-response manner (Fig 2C) but only the treatment with 32 μM of D-11 resulted in the death of the cells (S2B Fig). The results indicated that the increase of PI uptake was due to the inner membrane permeabilization by the sub-MIC D-11 treatment while at the MIC concentration fully membrane disruption can be achieved. Our results showed that D-11 does not only increase the permeability of the outer membrane of PAO1 but also increases the permeability of the inner membrane,

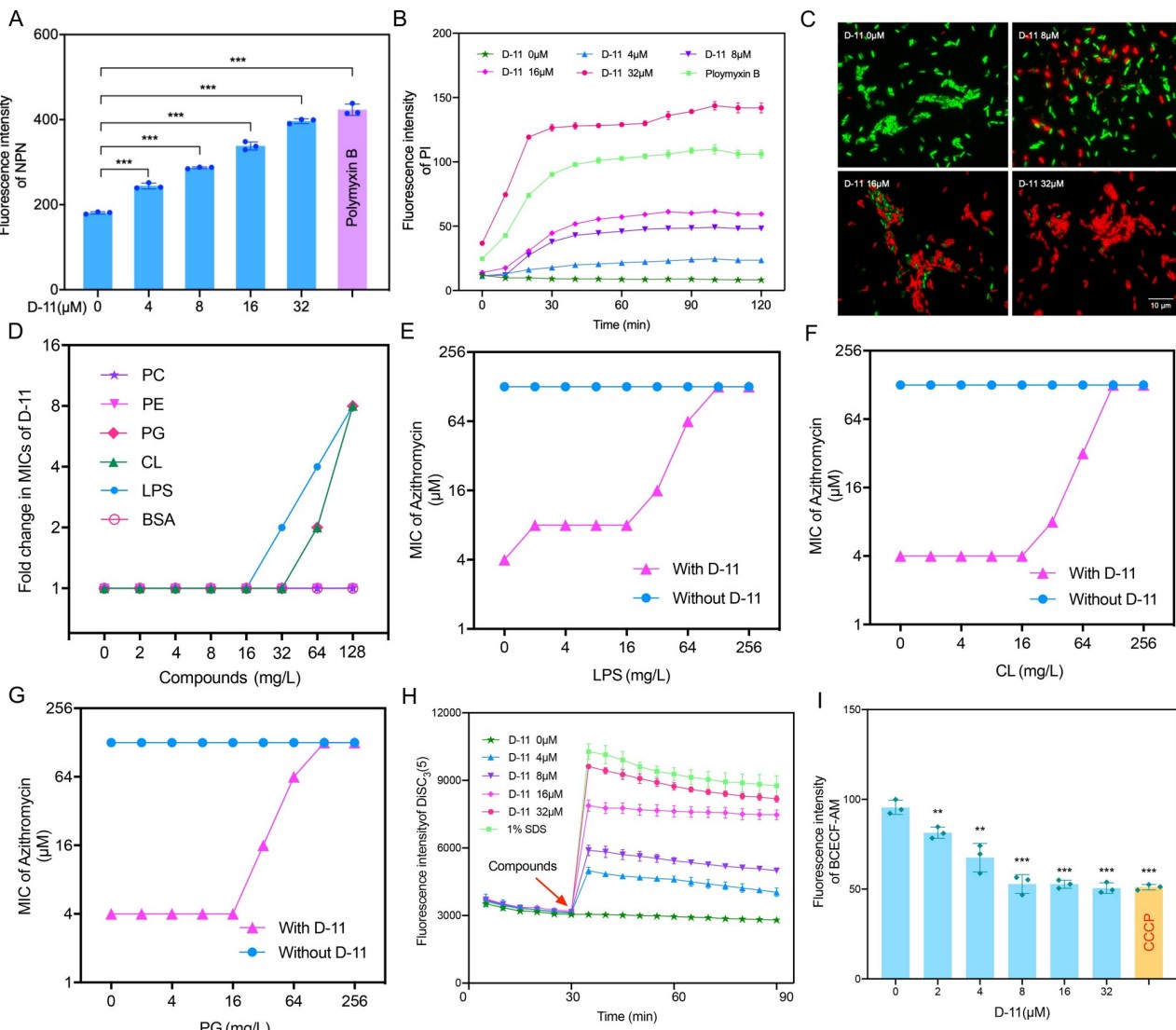

**Fig 2. D-11 increases membranes permeability, interacts with LPS and phospholipids and dissipates the membrane potential.** (**A**) D-11 dosage-related outer membrane permeability increment. Polymyxin B at 0.5mg/L was used as a positive control. (**B**) D-11 dosage-related inner membrane permeability increment. 0.5 mg/L polymyxin B was used as a positive control. (**C**) SYTO9 and PI live/dead cell staining of PAO1 treated with D-11. *P. aeruginosa* cells were treated with indicated concentrations of D-11 for 60 min. Cells were washed with PBS buffer and stained with SYTO9 and PI before being inspected with a microscope. (**D**) D-11 activity in the presence of BSA, PC, PE, PG, CL or LPS, the activity for the synergistic relation D-11/azithromycin in the presence of LPS (**E**), CL (**F**), and PG (**G**). D-11 lost the synergistic effect in the presence of LPS, CL or PG. The MIC of azithromycin was determined by checkerboard microdilution assays in the presence of LPS, CL or PG (0–256 mg/L) with 4 μM of D-11 or not. (**H**) DiSC3(5) membrane potential measurement for several D-11 dosages. Polymyxin B at 0.5mg/L was used as a positive control. The DiSC3(5) dye was incubated with PAO1 for 30 min followed by self-quenching and stabilization, then indicated concentrations of D-11 and SDS were added. The fluorescence units were monitored for 60 min. (**I**) PMF disruption measurement. CCCP was used as a positive control. All the tests were performed in triplicate and all the data are presented as mean ± SD. **, P < 0.01; ***, P < 0.001 by Student's t-test.

which indicates that D-11 may also interact with bacterial inner membrane components. To deepen our insight in the potential inner membrane target, we tested the antibacterial activity of the D-11 in the presence of Gram-negative bacterial compounds, i.e., LPS and three main phospholipid components of the membrane, including the zwitterionic phosphatidylethanolamine (PE, about 60% in PAO1), and the anionic lipids PG (about 21% in PAO1) and cardiolipin (CL, about 11% in PAO1) [33,34]. Among them, the mammalian cell

membrane component phosphatidylcholine (PC) and the negative charged non-bacterial related protein BSA were used as a control. Supplementation with exogenous LPS, PG or CL increased the MIC of D-11 by 8-fold, while PE, PC and BSA did not affect the MIC (Fig 2D). Next, we investigated the effects of these components on the synergistic effect of D-11 and azithromycin. Supplementation with exogenous LPS or other outer membrane stabilizers such as magnesium or calcium ions antagonized the synergistic effect of D-11 and azithromycin (Fig 2E and S3 Fig). Similarly, when PG or CL was added, D-11 lost the synergistic effect with azithromycin (Fig 2F and 2G), while PE or PC did not affect the synergistic effect (S3 Fig). Thus, D-11 may specifically interact with the Gram-negative bacteria-specific component LPS and the specific intrinsic phospholipids PG and CL.

## D-11 induces membrane depolarization, proton motive force dissipation

The increase in membrane permeability usually causes the dissipation of the membrane potential. So, we investigated the membrane potential of the cells treated with D-11 using the potential-sensitive membrane dye DiSC$_3$(5) [35]. A D-11 dose-related increment of the fluorescent signal indicated the membrane depolarization and the dissipation of the membrane potential (Fig 2H). Previous studies have shown that membrane depolarization is related to proton motive force (PMF) and the production of ROS [36,37]. The disruption of the PMF was detected by monitoring the fluorescence intensity of the BCECF-AM probe-labelled *P. aeruginosa* cells, treated with different doses of D-11. Carbonyl cyanide meta-chlorophenyl (CCCP), a compound known as proton uncoupler, was used as a positive control due to its ability to decrease the PMF (Fig 2I).

## D-11 reduces the intracellular ATP levels and causes NADH and ROS accumulation

Next, we measured the intracellular ATP levels, since the generation of ATP is the most relevant process to the membrane potential, which depends on the PMF generated by the electron transport chain (ETC). CCCP was used as a positive control due to its ability to decrease the intracellular ATP level. We found that the intracellular ATP level decreased rapidly with the increase of the D-11 dose (Fig 3A) which is consistent with the dissipation of the PMF.

In order to further investigate the mechanism of the decrease of ATP level in the bacteria under D-11 treatment, the NADH level of cells was monitored measuring the NADH reduction of resazurin to the fluorescent resorufin [38]. Interestingly, the level of NADH in the bacterial cells increased upon the D-11 treatment (Fig 3B), which is in contrast to the decrease in ATP levels we observed. To verify the level of NADH, we used a fluorescent quantitative method to determine the absolute content of NADH in cells after treatment with D-11 for two hours. The results confirmed that the content of NADH in cells increased with an increase of the D-11 dose (Fig 3C). The inhibition of the electron transfer and oxidative phosphorylation process is also involved in the intracellular accumulation of ROS, which is related to the death of the bacteria. Thus, we monitored the intracellular ROS level of bacteria upon D-11 treatment and found that the intracellular ROS levels increased significantly (Fig 3D). When the ROS scavenger N-acetylcysteine (NAC) was added to the sample with 16 μM D-11, the accumulation of ROS was diminished (Fig 3D). In fact, the supplement of NAC increased the bacterial survival rate following the treatment of D-11 and azithromycin (S4 Fig). Thus, D-11 triggers the accumulation of ROS, which correspondingly aggravates membrane damage to further reduce bacterial viability. These results indicate that D-11 can dissipate the PMF, reduces ATP production, and increases oxidative stress to cause cell death.

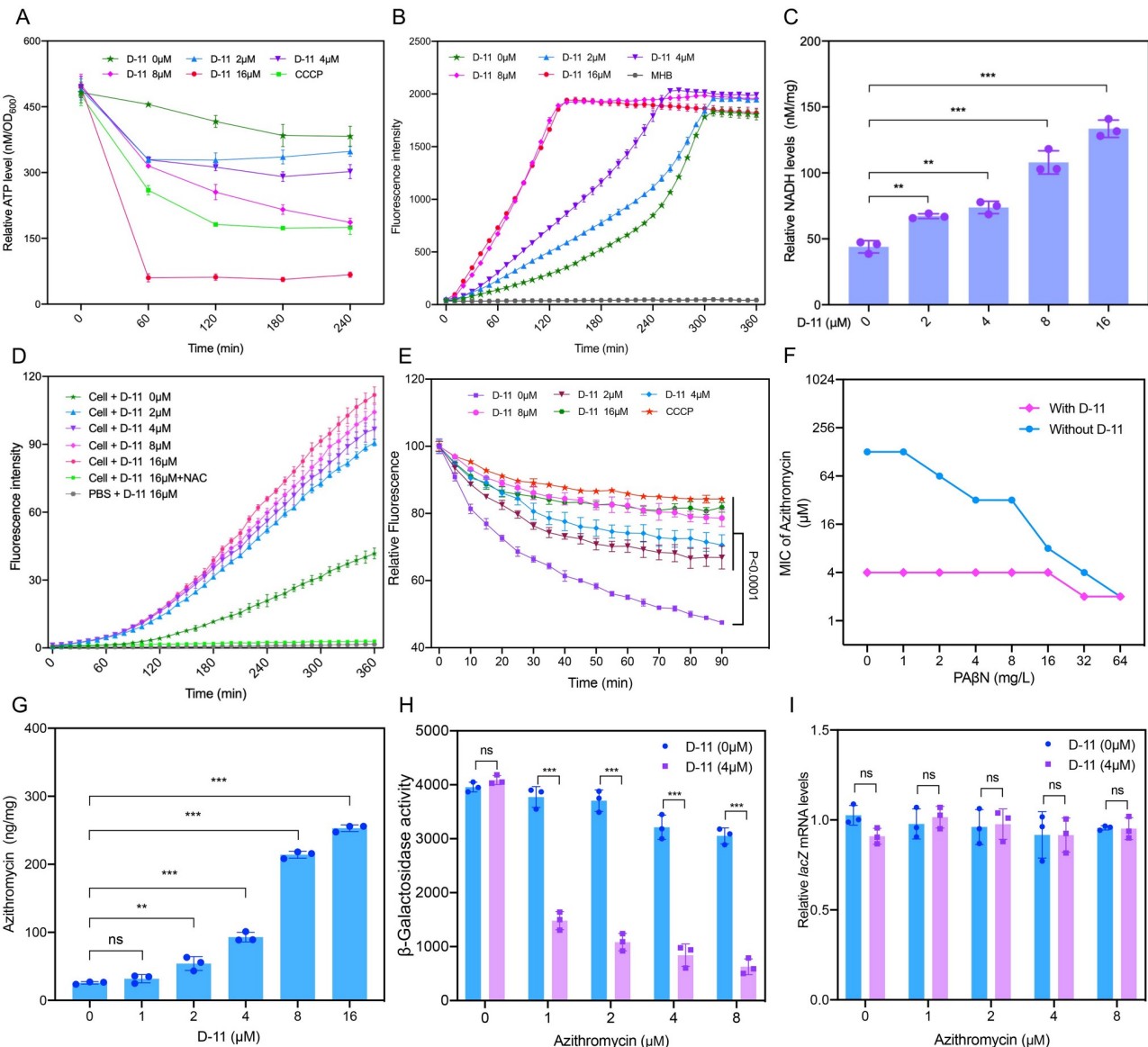

**Fig 3. D-11 impairs ATP and NADH levels, induces the accumulation of ROS, inhibits efflux pumps and promotes intracellular antibiotic accumulation.** (**A**) Intracellular ATP levels at different D-11 concentrations. 40 mg/L CCCP was used as a positive control. (**B**) Resazurin to resorufin reduction by NADH at different D-11 doses. (**C**) Intracellular NADH concentration after different D-11 doses. (**D**) Total ROS accumulation in PAO1 treated with D-11. Exogenous addition of NAC (6 mM) diminished the accumulation of ROS induced by 16 μM of D-11. (**E**) D-11 dosage-related EtBr efflux inhibition. PAO1 cells were incubated with EtBr in HEPES for 30 min at 37 ˚C then the cells were collected and resuspended in MHB medium containing the indicated concentrations of D-11 or 40mg/L CCCP. Then the fluorescence was monitored for 90 min at 37˚C. (**F**) D-11 antagonizes the effect of the efflux pumps inhibitor PAβN. (**G**) Intracellular D-11 dosage-related increment of azithromycin. (**H**) β-Galactosidase activity for the combination of azithromycin and D-11. (**I**) Relative *lacZ* mRNA expression levels for D-11 and D-11/azithromycin-treated cells. All the tests were performed in triplicate and all the data are presented as mean ± SD. *, P < 0.05; **, P < 0.01; ***, P < 0.001; ns, not significant by Student's t-test.

## D-11 promotes the intracellular accumulation of azithromycin

Sufficient accumulation of antibiotics in cells is a prerequisite for antibacterial activity, especially for Gram-negative pathogens [39,40]. Considering that the PMF is critical for the function of the efflux pumps [41], which play an important role in the resistance of macrolides in *P. aeruginosa*, we used ethidium bromide (EtBr) as a fluorescent probe to evaluate the activity

of the bacterial efflux pumps under D-11 treatment using CCCP as a positive control. A decreased efflux of EtBr in *P. aeruginosa* was observed after the incubation with D-11 (Fig 3E). This inhibitory effect of D-11 on efflux pumps was also confirmed by antagonizing the effect of the efflux pumps inhibitor PAβN (Fig 3F). In the presence of D-11, the MIC of azithromycin was only reduced by a factor of two with the higher concentration of PAβN, while the MIC of azithromycin was reduced 64-fold in the absence of D-11. This result indicates that the function of efflux pumps was already blocked in the presence of D-11.

The accumulation of azithromycin in *P. aeruginosa* in a dose-dependent manner was confirmed by the quantification of azithromycin inside the cell using an ELISA assay (Fig 3G). The target of azithromycin is the 50S ribosomal subunit, and the antibacterial mechanism is to inhibit protein translation. Next, we used a plasmid carrying a *lacZ* gene driven by an exogenous P$_{lac}$ promoter to determine the translation level of the β-galactosidase upon treatment with azithromycin. The β-galactosidase enzyme activity levels were decreased significantly in the presence of D-11, while the mRNA levels were similar with or without D-11 (Fig 3H and 3I). All these data suggest that the synergistic effect of macrolides with D-11 is not exclusively related to the increased bacterial membrane permeability but also to the accumulation of these antibiotics within the cells due to attenuation of the efflux pumps.

## The response of *P. aeruginosa* to D-11 and azithromycin

In order to further understand the mechanism of the synergy of D-11 and azithromycin, we performed transcriptome analyses. The comparison of treatment with the combination of 4 μM D-11 and 8 μM azithromycin to untreated cells revealed an up-regulation of 1715 and downregulation of 1683 genes, while 171 genes were downregulated, and 300 genes upregulated when treated with 4 μM D-11 alone and 1075 genes downregulated, and 1198 genes upregulated when treated with 8 μM azithromycin alone (S2A–S2C Table).

The resistance of *P. aeruginosa* to polycationic AMPs is believed to be mainly dependent on the modification of LPS (addition of 4-amino-4-L-deoxyarabinose (Ara4N) to the phosphate groups of LPS mediated by the *arn* operon (*arnBCADTEF*) [42]. When exposed to low concentrations of D-11 (4 μM), the *arn* operon and the two-component system *pmrAB* involved in its regulation and the magnesium transporter encode genes *mgtA/E* were significantly upregulated (Fig 3A). When exposed to the combination of D-11 and azithromycin, oxidative phosphorylation related genes including ATP synthase, NADH-quinone oxidoreductase and cytochrome oxidase were significantly downregulated (Fig 4A). The damage of the oxidative phosphorylation process is involved in the intracellular accumulation of ROS. At the same time, treatment with azithromycin or the combination significantly reduced the expression of major oxidative stress-related enzymes (KatA, SodB, AhpC) (Fig 4A), thus further increasing the synergistic effect. Moreover, glycolysis, TCA cycle and fatty acid oxidation-related genes were significantly down-regulated (Fig 4A), which also indicates changes in metabolic levels. The decrease in the metabolic fitness level may be due to the obstruction of oxidative phosphorylation and the accumulation of NADH, which reduce metabolic activity through feedback effects. In *P. aeruginosa*, the TpiA-mediated conversion between glyceraldehyde-3-phosphate (G3P) to dihydroxyacetone phosphate (DHAP) serves as a critical link between glycolysis and phospholipid synthesis [43]. The high expression of *tpiA*, *gpsA*, *pgpA* and *cls* also indicates the enhancement of phospholipid synthesis activity and the decrease of glucose metabolism. However, these gene expression changes were not observed in the case of D-11 treatment alone. We speculate that the enhancement of the membrane repair function under the treatment of a low concentration of D-11 might prevent enough peptides to reach the bacterial inner membrane to function (the MIC of D-11 alone is 32 μM). In the case of combined

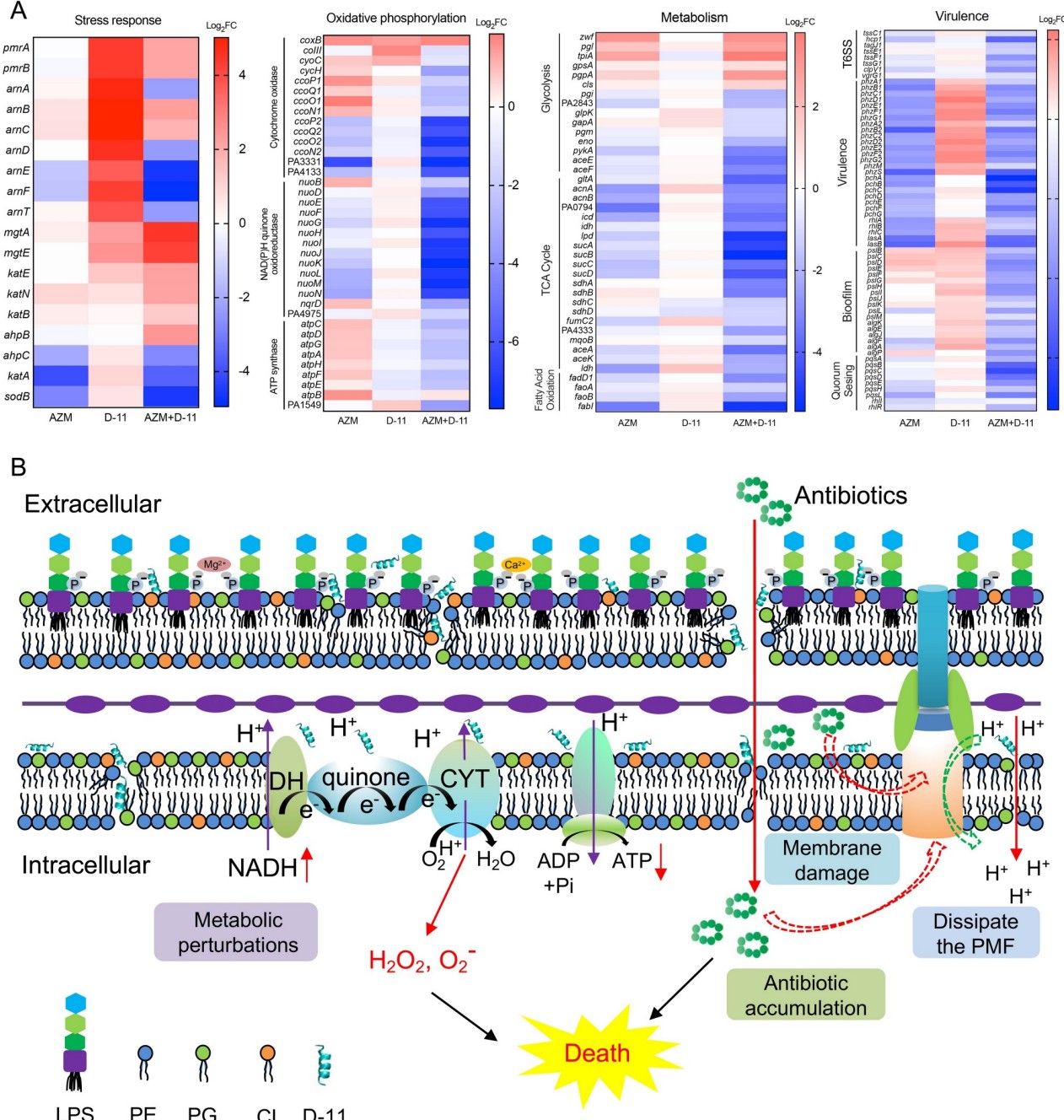

**Fig 4. Transcriptome analysis and proposed synergy action model.** (**A**) The effect of D-11 alone, azithromycin alone or the combination over the expression of several genes involved in stress response, oxidative phosphorylation, metabolism and virulence. AZM, 8 μM azithromycin treatment alone; D-11, 4 μM D-11 treatment alone; AZM + D-11, the combination of azithromycin and D-11 treatment. (**B**) Proposed action mechanism for the synergy of D-11 in combination with antibiotics. D-11 restores the susceptibility of *P. aeruginosa* to antibiotics through membrane-mediated mechanisms by interacting with LPS, CL and PG. These reactions trigger membrane depolarization and induce membrane dysfunction, further resulting in metabolic perturbations and ROS accumulation. In addition, more antibiotics rapidly accumulate inside bacteria due to the increased membrane permeability and dysfunction of efflux pumps mediated by the dissipation of membrane potential. The accumulation of ROS and antibiotics working together resulted in the death of cells.

treatment, the membrane repair function is inhibited, and more D-11 can come into contact with the inner membrane of bacteria to exert its function. At the same time, the ability of azithromycin to inhibit the expression of virulence factors of *P. aeruginosa* under the combined treatment conditions was also enhanced (Fig 4A).

These results together demonstrate that D-11 specifically potentiates azithromycin activity via the increased accumulation of intracellular antibiotics by combined binding with LPS and membrane phospholipids, resulting in membrane damage, inhibition of the ETC, and dissipating the membrane potential. As a result, the efflux pump function is inhibited, and intracellular ROS accumulates to further enhance the efficacy of killing the bacteria (Fig 4B).

## D-11 enhances azithromycin activity on biofilms

Bacterial biofilm formation is a well-known virulence mechanism that provides strong resistance to antibiotics and is involved in causing infection relapses [44]. *P. aeruginosa* biofilms are particularly difficult to be eliminated because bacterial cells embedded in biofilms can reach up to 1,000 times more resistant to antibiotics [8], and they are responsible for relapses in the infections [45,46]. It is well known that although azithromycin is inactive against free *Pseudomonas* cells, its activity against biofilm growing cells and on the bacterial virulence is well documented [47]. Therefore, we analyzed whether D-11 can enhance the ability of azithromycin to control biofilms. We first tested the ability of azithromycin on the prevention of biofilm formation in PAO1, either in the presence or absence of D-11. As shown in Fig 5A and 5B and S5 Fig, the inhibitory effect of azithromycin on biofilm formation is significantly enhanced in the presence of D-11. Azithromycin exhibits anti-biofilm activity against *P. aeruginosa* by inhibiting the expression of biofilm formation related genes and the quorum sensing (QS) systems [48]. Our transcriptomic results revealed that the expression of biofilm formation and QS related genes was highly repressed by the treatment of the combination (Fig 4A). Therefore, suppression of biofilm formation related genes contributes to the ability of azithromycin to inhibit biofilm formation. Next, we tested the effect of D-11 on the ability of azithromycin to kill bacteria in the already established biofilms. The combination of D-11 and azithromycin significantly enhanced the killing efficacy on the biofilm-associated cells (Fig 5C). To confirm the killing of biofilm-associated cells, we tested the number of viable planktonic bacteria in each tube after the treatment. The results showed that the bacterial viability was reduced by about 99% in the supernatant of the co-treatment group (Fig 5D).

## D-11 has a synergistic effect with multiple antibiotics

Our results so far demonstrated that D-11 not only significantly increases the permeability of the outer membrane of *P. aeruginosa*, but also significantly inhibits the activity of the efflux pumps, so we speculated that it may also have a synergistic effect with other antibiotics. Therefore, we tested the synergistic effect of D-11 with 42 other drugs. Among them, 25 antibiotics showed strong synergistic effects with D-11 (Fig 6A and S3 Table). It is worth noting that the MICs of the rifamycin family antibiotics were reduced by 512-2048-fold, and the MICs for colistin related antibiotics were reduced by 256-fold in the presence of 4 μM of D-11. Good synergistic effects were also observed for tetracycline and quinolone related antibiotics (Fig 6A). In addition, the MICs for antibiotics not yet used in the clinic such as the aminocoumarins coumermycin A1 or novobiocin were reduced 512-fold (Fig 6A). Only a few antagonistic effects were observed with aminoglycosides given that the uptake of aminoglycoside antibiotics is highly dependent on the bacterial membrane potential [49]. These results indicate that D-11

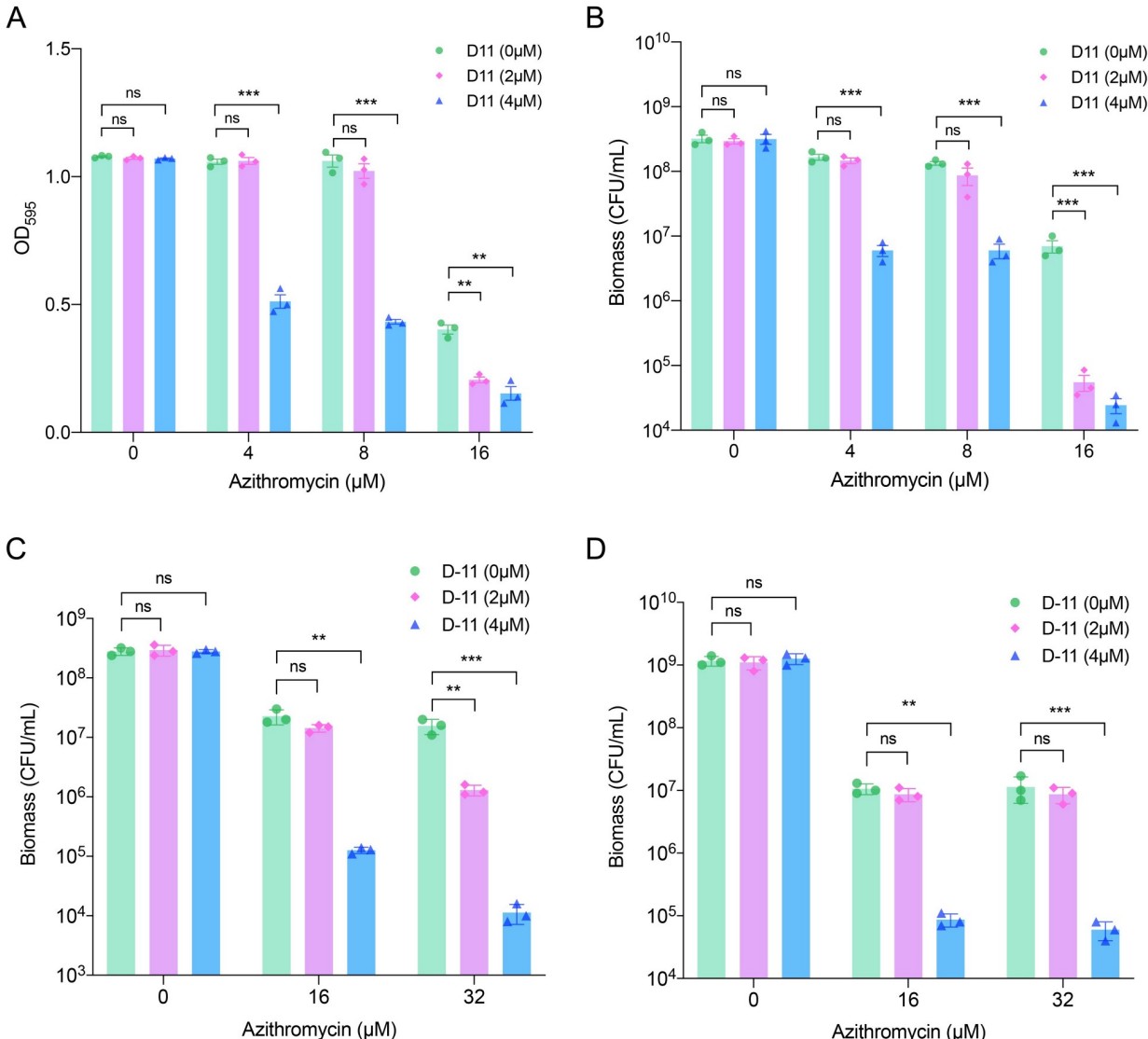

**Fig 5. D-11 increase the anti-biofilm ability of azithromycin.** (**A**) The biofilm was grown at the indicated conditions and stained with crystal violet. The crystal violet was dissolved in ethanol and measured at a wavelength of 595 nm. (**B**) The biofilm was grown at the indicated conditions and dispersed by ultrasound. The bacterial count inside of biofilm was determined by plating. The established biofilm was treated with indicated conditions for 24 hours. The bacterial count inside the biofilm (**C**) or in the supernatant (**D**) was determined by plating. All the tests were performed in triplicate and all the data are presented as mean ± SEM. **, P < 0.01; ***, P < 0.001; ns, not significant by Student's t-test.

can be used as an effective broad-spectrum antibiotic adjuvant to enhance the antibacterial effect of multiple antibiotics against *P. aeruginosa*.

To verify the universality of these synergistic effects, we tested these active combinations on three other *P. aeruginosa* reference strains and nine clinical isolates in the presence of 4 μM D-11. For the macrolides, some additive effects were observed in spiramycin and telithromycin (Fig 6B and S4 Table). All the tetracyclines and rifampicins showed synergistic effects in most of the cases against all of the strains besides AUMC-Pa-5, on which only a few antibiotics showed synergy (Fig 6B and S4 Table). Quinolones and β-lactams only show synergistic effects in a small number of cases. While polymyxin B, chloramphenicol and aminocoumarins showed good synergy in most of the cases (Fig 6B and S4 Table).

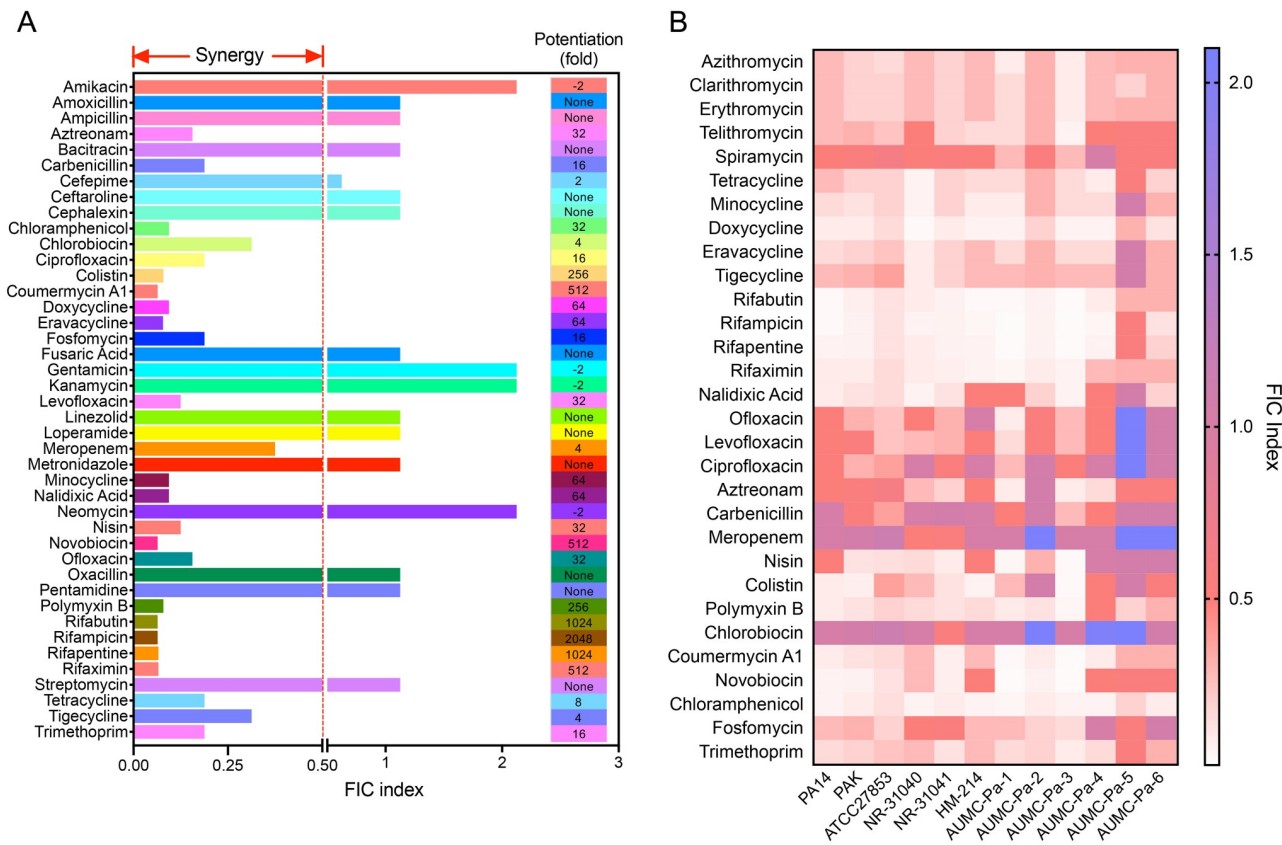

**Fig 6. D-11 enhances the activity of multiple antibiotics.** (**A**) synergistic effects of D-11 with multiple antibiotics. The FICI values of D-11 combined with multiple antibiotics and the fold change of MICs of those antibiotics in the presence of 4 μM D-11. (**B**) The heat map of the FIC indices for the synergistic effects of multiple antimicrobial agents combined with 4 μM D-11 against three standard *P. aeruginosa* strain and nine clinical isolates. Synergy is indicated by an FIC index ≤ 0.5.

## D-11 increases the antimicrobial activity of several antibiotics in blood

Although we have observed significant synergistic effects of D-11 and several antibiotics *in vitro*, considering the complex environment of bacterial infections, it remains unclear whether this synergistic effect can play a role in real infections. We developed an *ex vivo* bacteremia infection model through whole blood infection with $5 \times 10^7$ CFU/mL of *P. aeruginosa* and quantify the progression of infection in the blood by detecting the residual number of bacteria after four hours of treatment under different conditions. As shown in Fig 7A, in the presence of D-11, the antibacterial abilities of those antibiotics in the blood were significantly enhanced, while D-11 or antibiotics alone cannot control the bacterial infections (Fig 7A and S6 Fig). Among them, the combination of chloramphenicol, doxycycline and rifampicin with D-11 showed the best antibacterial effects in blood, similar to the macrolides after four hours of treatment (Fig 7A). However, polymyxin B, novobiocin and coumermycin A1 showed relative week antibacterial effects in the blood (S6 Fig). Notably, azithromycin, clarithromycin, chloramphenicol and doxycycline can control the infection even more than 24 hours in the presence of 4 μM D-11 (S7 Fig).

## The efficacy of D-11 in a murine model

Encouraged by the *in vitro* and *ex vivo* results and the therapeutic potential of the combinations to reduce drug doses and toxicity, we tested the efficacies of D-11 combined with the

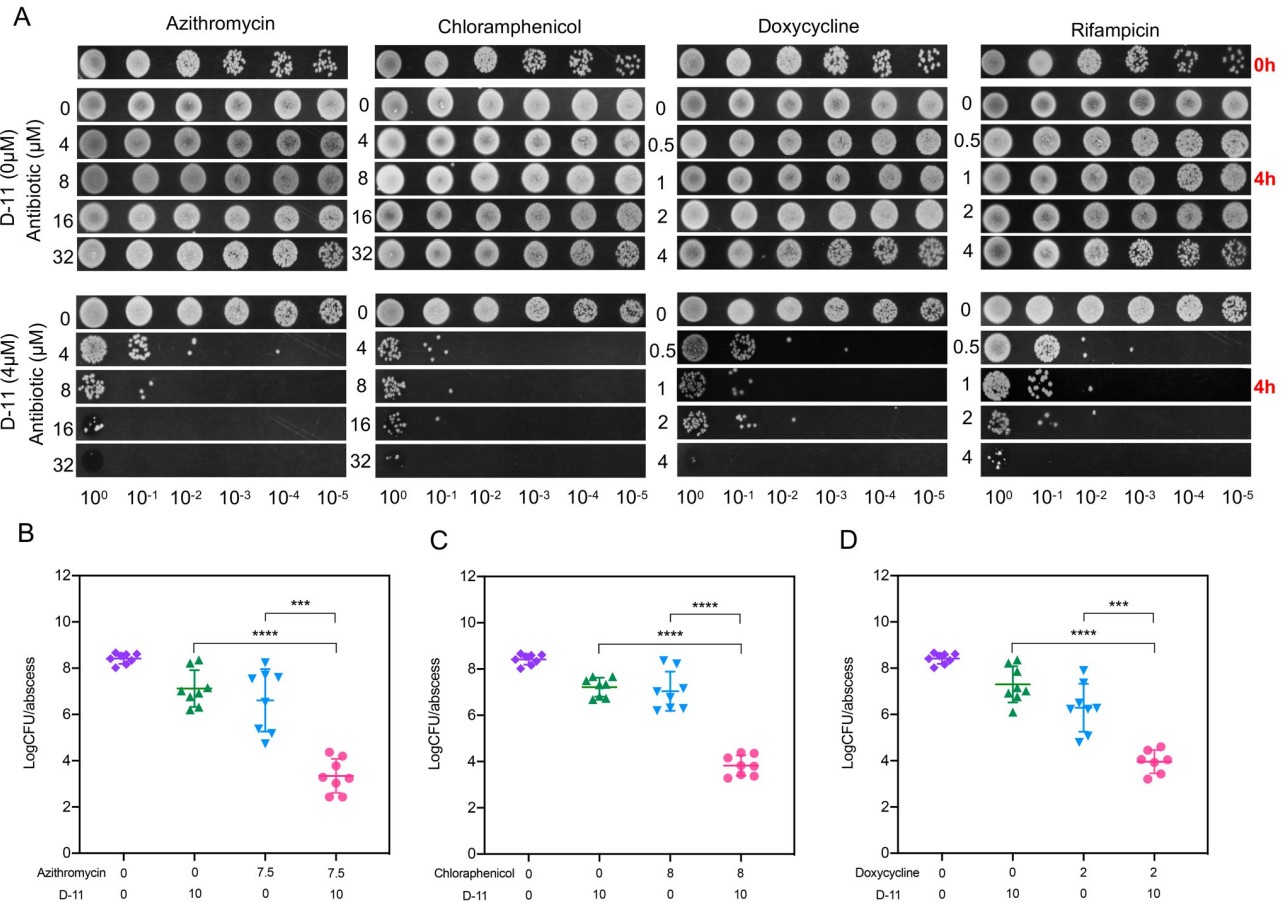

**Fig 7. D-11 enhances the activities of multiple antibiotics *ex vivo* and *in vivo*.** (**A**) *Ex vivo* bacteremia model for antibiotics, D-11, and their combinations antimicrobial activity. 10 μL of serial decimal dilutions of each one of the combinations of antibiotics with/without D-11 (4 μM) tested in blood were dropped. *In vivo* efficacy of azithromycin (7.5 mg/kg) (**B**), chloramphenicol (8 mg/kg) (**C**), doxycycline (2 mg/kg) (**D**) and D-11 (10 mg/kg) alone or their combination against *P. aeruginosa* PAO1 (n = 7 for doxycycline and D-11, n = 8 for the other groups). All the three experiments are using the same group of negative control which treatment with saline. All the data are presented as mean ± SD. ***, P < 0.001; ****, P < 0.0001; ns by Student's t test.

antibiotics in a murine model of *P. aeruginosa* infection. In a murine abscess infection model, we observed that a single dose of the combinations of D-11 (10 mg/kg) with azithromycin (7.5 mg/kg), chloramphenicol (8 mg/kg) and doxycycline (2 mg/kg) were highly efficacious, causing more than 2–3 $\log_{10}$ reductions of CFU in the abscess compared to each individual antibiotic (Fig 7B, 7C and 7D). Interestingly, the mechanism of these drugs exhibiting the best synergistic effects with D-11 are all involved in the inhibition of protein synthesis, indicating that D-11 is more likely to have a better synergistic effect with drugs related to protein synthesis inhibition. This finding demonstrates the potential of D-11 as an antibiotic adjuvant to potentiate these antibiotics against *P. aeruginosa in vivo*.

## Discussion

Activity against multiple targets, a unique action mechanism, no cross-resistance development with established antimicrobials, stability and activity in human fluids, no toxicity and specificity towards bacteria constitute the most valuable characteristics of newly designed antimicrobials [4]. Here we have explored the combined effects of selected and newly designed peptides

with macrolides-related antibiotics against *P. aeruginosa*. Our data support that initially, D-11 affects the integrity of the outer membrane, which promotes the accessibility of antibiotics to their target. *P. aeruginosa* possesses the most antibiotic-restrictive outer membrane among the Gram-negative bacteria. It is known that polycationic molecules, such as polymyxins, cationic AMPs or divalent cation chelators can act as outer membrane permeabilizing agents [50,51]. In fact, in the case of *P. aeruginosa* previous studies have shown that EDTA, cationic peptides, polymyxin B nonapeptide, lysine polymers, and protamine can enhance the bacterial sensitivity to different antibiotics [52], albeit with less efficiency or more toxicity than with D-11 in our present study.

Antimicrobial peptides derived from host defense proteins have been proposed as a new strategy against Gram-negative bacterial infections. Previous studies have shown that serval cationic peptides have a synergistic effect with carbenicillin, ciprofloxacin and nalidixic acid in *P. aeruginosa* [53]. IDR-1018 also has a synergistic function with a variety of antibiotics to remove the biofilm formed by *P. aeruginosa* or *Staphylococcus aureus* [54]. When used in combination with ceftazidime, EDC34 can effectively improve the survival rate of mice infected with *E. coli* and *P. aeruginosa* [55]. SLAP-S25 effectively enhances the activity of colistin, vancomycin and rifampin against MDR *E. coli* related infections [31]. hLF1-11 is an 11-amino acid peptide derivative of human protein lactoferrin, which shows a similar ability to clear infection as gentamicin in the rabbit osteomyelitis infection model. The safety of this peptide has been tested in phase I clinical trials [56,57]. In this study, we demonstrated the enhanced therapeutic efficacies of the combination of D-11 with various antibiotics against *P. aeruginosa* *in vitro* and *in vivo*, which indicates that D-11 can be used as an effective broad-spectrum antibiotic adjuvant to enhance the antibacterial effect of multiple antibiotics against *P. aeruginosa*.

The positively charged amino acids in AMPs can interact with the anionic moieties of the plasma membrane phospholipids, which also contributes to the antimicrobial effect of AMPs [34]. For example, AMPR-11 specifically binds lipid A and CL [58], SLAP-S25 triggers membrane damage by binding to LPS in the outer membrane and PG [31]. Since D-11 contains five positively charged amino acids, it is easier to bind to bacterial negatively charged membrane components. In fact, our results show that exogenous LPS, PG and CL block the function of D-11 suggesting an interaction between them. Interestingly, no interaction with PE or with the mammalian cells phospholipid PC was observed. Cationic peptides like D-11 are usually non-toxic to cell lines, but they are commonly quite hemolytic and have poor stability in plasma. Previously we have shown that D-11 is stable in plasma, and even high-concentration of D-11 shows no toxicity to HEK-293 cells or hemolytic activity [14]. The absence of activity against eukaryotic cells reported could be related not only to charge repulsion (eukaryotic cells are usually positively charged) but also to the absence of targets for this peptide such as membranes and could also indicate specificity towards bacteria. However, there were still some concerns on the clinical immunogenicity of the D-amino acid peptide. In the *in vivo* test, no obvious side effects were observed at the concentration of 10 mg/kg D-11. However, we found that subcutaneous injection of 20 mg/kg of D-11 was slightly irritating, while high-dose (50 mg/kg) D-11 resulted in strong irritation and caused skin tissue damage 24 hours post injection.

The activity of D-11 is not restricted to membrane damage but also to an alternative action mechanism in the dissipation of PMF which inhibits the activity of multi-drug efflux pumps (an established Gram-negative drug resistance mechanism). We speculate that the peptide quickly binds to LPS disrupting the outer membrane. As a result, D-11 can reach the inner membrane, which is also damaged, depolarizing the cells and impairing the proton motive force and the membrane-bound aerobic respiration electron transport chain inducing NADH accumulation, which also produces toxic effects [59]. Subsequently, superoxide ($O_2^-$) is

released from the complex [60]. A previous study has shown that the mechanism of the cationic antibacterial peptide polymyxin against Gram-negative bacteria partly involves the inhibition of NADH-quinone oxidoreductase activity [61]. The increase in NADH levels indicates likely damage of electron transport and oxidative phosphorylation, which may result in the inhibition of NADH-quinone oxidoreductase activity. The inhibition of NADH-quinone oxidoreductase activity and the increased inner membrane permeability results in the dissipation of the PMF and reduces the generation of ATP. Another study has reported that cathelicidin peptides restrict bacterial growth via membrane perturbation and induction of ROS [62]. In addition, host immune antimicrobial LL-37 disrupts the proper flow of electrons through the electron transport chain, releasing oxidative species into the periplasm [63]. The ROS induced by D-11 may interfere with a variety of molecules, such as lipids, cytoplasmic proteins and DNA, and ultimately promote the killing of bacteria by antibiotics [62]. As a result, both ATP synthesis and efflux pumps are impaired, promoting the intracellular accumulation of drugs, which can also enter more easily due to membrane alterations. Because of these malfunctions, the bacteria are largely disarmed against the biocidal effect of antimicrobials. This hypothesis was fully supported by the data obtained following the transcriptomic analyses of the cells after the treatments.

In summary, we have carried out a detailed study on the action mechanism of D-11, which has a strong ability to increase membrane permeability by inhibiting the multidrug efflux pumps and rendering *P. aeruginosa* more susceptible to the antimicrobial activity of macrolides *in vitro* and mouse infection models. D-11 also showed a strong synergistic effect with a variety of antibiotics and displays satisfactory safety, indicating that D-11 could be used as an antibiotic adjuvant for the treatment of clinical *Pseudomonas* and possibly also other Gram-negative pathogens.

## Materials and methods

### Ethics statement

The animal infection experiments described in this study were performed following the National and Nankai University guidelines on the use of animals in research. The protocol with the permit number NK-04-2012 was approved by the animal care and use committee of the College of Life Sciences, Nankai University.

### Bacterial strains, primers, plasmids and antibiotics

The bacterial strains, primers and plasmids used in this study are listed in S5 Table. The antibiotics susceptibilities of the clinical isolates were listed in S6 Table. Bacteria were cultured in LB medium or cation-adjusted Muller-Hinton broth (CA-MHB) at 37 ˚C with agitation at 220 rpm. All the antibiotics and chemical compounds used in this study were purchased from Sigma-Aldrich. The peptides used in this work were synthesized by Pepscan (Lelystad, The Netherland) and all of the peptides had a purity higher than 95%.

### MIC determination and synergy test

The MIC tests were performed in triplicate using broth microdilution in accordance with Clinical and Laboratory Standards Institute (CLSI) recommendations (http://iacld.ir/DL/public/CLSI-2018-M100-S28.pdf). Briefly, drugs were two-fold diluted in MHB and mixed with an equal volume of bacterial suspensions containing approximately $1 \times 10^6$ CFU/mL in 96-well microliter plate. After 18 h incubation at 37 ˚C, the MIC values were defined as the lowest concentrations of antibiotics with no visible growth of bacteria. For the initial screening of

synergistic effect, 8 μM antimicrobial peptide was used in combination with azithromycin if the MIC of the peptide is higher than 32 μM. Otherwise, 1/4 MIC of the peptide was used in the combination. For effective compounds, a standard checkerboard broth microdilution test was performed to test the synergy of the combined antimicrobial agent as previously described [64]. If needed, membrane compounds PG, PE, PC, CL or LPS, or membrane stabilizers $MgCl_2$ or $CaCl_2$, or efflux pumps inhibitor PAβN were added to the broth medium to clarify the effect of other substances on the synergy between peptide and antibiotics. All the tests were performed in triplicate.

The fractional inhibitory concentration (FIC) indices were calculated using the previously described [64] formula FICI = FICa + FICb = MICac/MICa + MICbc/MICb. The FICI was interpreted according to EUCAST [65] as follows: synergistic, FICI≤0.5; additive, 0.5<FICI≤1; indifferent, 1<FICI<2; antagonistic, FICI ≥2. For the FICI calculations, twice the highest concentration tested was used in the cases where the MIC was not reached.

### Assessment of outer membrane permeability

The integrity of the outer membranes was analyzed with the fluorescent probe 1-N-phenyl-naphthylamine (NPN, Sigma-Aldrich). Briefly, bacteria cultured overnight were diluted 1: 100 into fresh CA-MHB and cultured at 37 ˚C to the late log phase ($OD_{600}$ = 1). The cells were washed with a 5 mM HEPES buffer containing 5 mM glucose (GHEPES). The bacterial suspension was standardized to an $OD_{600}$ = 0.5 in GHEPES buffer. NPN was added to the cells at a final concentration of 30 μM. The D-11 was added into the bacterial suspension at the indicated concentrations. The fluorescence was monitored at excitation/emission of 350/420 nm every 10 min for 1 hour with a luminometer (Varioskan Flash; Thermo Scientific). Polymyxin B (1×MIC, 0.5 mg/L) treatment was used as a positive control. All the tests were performed in triplicate.

### Membrane integrity assay

Overnight bacterial cultures were diluted 1:100 in fresh CA-MHB medium and grown at 37˚C to an $OD_{600}$ of 1. The cells were washed three times with PBS buffer and adjusted the $OD_{600}$ = 0.5, followed by the addition of 1 μM of PI (Thermo Fisher Scientific) in the presence of D-11 or polymyxin B (1×MIC, 0.5 mg/L). The fluorescence was monitored at an excitation/emission wavelength of 535/615 nm every 10 min for 2 hours at 37˚C with a luminometer (Varioskan Flash; Thermo Scientific). All the tests were performed in triplicate.

### Bacterial viability assay

The LIVE/DEAD BacLight bacterial viability kit (Invitrogen) was used to evaluate the dead bacteria induced by D-11. Overnight bacterial cultures were diluted 1:100 in fresh CA-MHB medium and grown at 37 ˚C to an $OD_{600}$ of 1. The cells were washed three times and resuspended in PBS. The bacteria were treated with different concentrations of D-11 at 37˚C for 1 hour. Subsequently, the bacteria were collected, washed three times with PBS and resuspended in PBS. Two different dyes (20 mM propidium iodide and 3.34 mM SYTO9) were added at a ratio of 1:1 (v/v) and incubated for 30 min in dark at room temperature. 2 μL of the sample were mounted on a 1.5% agarose pad, and then imaged using a Nikon Ti-E microscope (Nikon Instruments, Tokyo, Japan) equipped with a Hamamatsu Orca Flash 4.0 camera.

### Membrane potential assay

The bacteria were cultured at 37˚C in CA-MHB medium to an $OD_{600}$ of 1. The cells were washed three times with 5mM GHEPES buffer. The bacterial suspension was standardized to

an $OD_{600} = 0.5$ in GHEPES buffer. 3,3-Dipropylthiadicarbocyanine iodide DiSC3(5) was added to the cells at a final concentration of 2 μM. The D-11 peptide was added into the bacterial suspension with the indicated concentration. The fluorescence was monitored at excitation/emission of 622/670 nm every 10 min for 2 hours with a luminometer (Varioskan Flash; Thermo Scientific). SDS (1%) treatment was used as a positive control. All the tests were performed in triplicate.

## Proton motive force disruption assay

The PMF assay was performed as previously described with minor modifications [66]. Overnight cultured PAO1 cells were collected via centrifugation and washed with 50 mM potassium phosphate buffer (pH 6) and potassium phosphate buffer (50 mM, pH 6) containing EDTA (5 mM), then the cells were re-suspended in 1 mL the same phosphate-EDTA buffer. BCECF-AM (2 mM, 10 μL) was added and incubated for 1 hour at room temperature. The cells were concentrated to 120 μL in the phosphate-EDTA buffer and incubated on ice for 4 hours. An aliquot of the cell suspension (2 μL) was then added to fresh phosphate-EDTA buffer (200 μL) with the indicated concentration of D-11 or CCCP (40 μg/mL) as a positive control. Fluorescence was monitored at excitation/emission of 500/522 nm with a luminometer (Varioskan Flash; Thermo Scientific). The data showed are representative of results from three technical replicates.

## Bacterial ATP level assay

The intracellular ATP levels were determined using a BacTiter-Glo Microbial Cell Viability Assay (Promega) kit. Bacteria were cultured in CA-MHB medium at 37˚C until the late-logarithmic phase ($OD_{600} = 1$). Cells were treated with the indicated concentration of D-11 at 37 ˚C. 100 μL of cell culture were taken out, mixed with 100 μL BacTiter-Glo reagent every hour and incubated for 5 minutes at room temperature. Luminescence was measured with a luminometer (Varioskan Flash; Thermo Scientific). CCCP (40 μg/mL) was used as a positive control. The ATP concentration was calculated using a standard curve made with a commercial ATP solution. The $OD_{600}$ was determined before ATP levels were measured. Relative ATP levels were calculated by the $OD_{600}$. All the tests were performed in triplicate.

## Resazurin assay

The bacteria were cultured in the CA-MHB medium at 37˚C to an $OD_{600}$ of 1. D-11 was added into the medium at the indicated concentrations. Resazurin was added at a final concentration of 0.1 mg/ml. Fluorescence resorufin was monitored at excitation/emission of 550/590 nm every 10 min throughout 6 h with a luminometer (Varioskan Flash; Thermo Scientific). All the tests were performed in triplicate.

## Bacterial NADH level assay

Bacterial NADH levels were determined using an Amplite fluorimetric NAD/NADH ratio assay kit (AAT Bioquest, Inc., USA). The bacteria were cultured in CA-MHB medium at 37˚C to an $OD_{600}$ of 1. The indicated concentration of D-11 was added to treat the bacteria for 1 hour, the bacteria were collected and suspended in PBS and disrupted by ultrasonic treatment. The supernatant was collected by centrifugation at 12000 g for 10 minutes at 4˚C. The NADH level was measured according to the manufacturer's instructions. The standard curve was prepared using the standard NADH provided by the kit. The total protein level was quantified by a BCA analysis kit. All the tests were performed in triplicate.

## ROS measurement

Bacterial intracellular ROS levels were determined using a Fluorometric Intracellular Ros Kit (Sigma-Aldrich, Catalog Number, MAK143). The bacteria were cultured in CA-MHB medium at 37˚C to an $OD_{600}$ of 1. The bacteria were then treated with D-11 at the indicated concentrations. The ROS level was measured according to the manufacturer's instructions. The antioxidant NAC (6 mM) was used as a control to neutralize the production of ROS. All the tests were performed in triplicate.

## EtBr efflux assay

Overnight bacterial cultures were diluted 1:100 in fresh CA-MHB medium and grown at 37 ˚C to an $OD_{600}$ of 1. The cells were washed three times with 10 mM HEPES buffer and adjusted the $OD_{600} = 0.5$, followed by the addition of 5 μM EtBr in the presence of D-11 or 40 mg/L CCCP and incubated 30 min at 37 ˚C. The cell was collected by centrifugation and resuspended in MHB medium containing the indicated concentration of D-11 or 40 mg/L CCCP. Then the fluorescence was monitored at excitation/emission of 525/600 nm every 5 min for 90 min at 37˚C with a luminometer (Varioskan Flash; Thermo Scientific). All the tests were performed in triplicate.

## Azithromycin uptake assay

To quantify the uptake of the azithromycin, bacteria were grown in MHB to an $OD_{600}$ of 1 and then incubated with 8 mg/L azithromycin and the indicated concentrations of D-11 for 1 hour. The bacteria were collected by centrifugation and washed three times with PBS. The bacterial cells were lysed in PBS by sonication. The cellular azithromycin levels were determined using an Azithromycin ELISA Kit (Lvshiyuan, Shenzhen, China). Total bacterial protein concentrations were quantified using a BCA analysis for calibration.

## β-galactosidase assay

Bacterial cells were grown in LB at 37 ˚C until the $OD_{600}$ reached 1. 0.5 ml of the bacterial culture was collected and resuspended in 1.5 ml of Z buffer (50 mM β-mercaptoethanol, 60 mM $Na_2HPO_4$, 60 mM $NaH_2PO_4$, 10 mM KCl, 1 mM $MgSO_4$, pH 7.0). The β-galactosidase activity was determined as previously described [67].

## RNA isolation and Real-Time PCR (RT-PCR)

Bacterial cells were cultured under the indicated conditions. 1 mL bacteria were collected and resuspended in 500 μL TRIzol reagent (Life technologies). Total RNA was extracted by chloroform extraction and isopropanol precipitation. Residual DNA was digested with RNase free recombinant DNase I (Roche). RNA is dissolved in RNase free water. cDNAs were synthesized using random primers and reverse transcriptase (Invitrogen). RT-PCR was performed with the SYBR II Green supermix (BioRed). The ribosomal gene *rpsL* was used as the internal control [68]. All the tests were performed in triplicate.

## Transcriptome analysis

Overnight bacteria cultures were diluted 1:100 in fresh CA-MHB medium and grown at 37 ˚C to an OD600 of 0.4. Subsequently, the bacterial were treated with 4 μM D-11 or 8 μM azithromycin or the combination for 1 hour (two replications per treatment), 1 mL of culture was collected for RNA isolation by TRIzol reagent. The Zymo-Seq RiboFree Total RNA Library Kit (Zymo Research, USA) was used to prepare the library preps for Illumina sequencing using

800ng of the extracted total RNA. Samples were sequenced on the Illumina NexSeq 500 to generate 75 bases singel end reads (75SE) with an average read depth of 12M reads per sample. The quality of the resulting fastq reads we checked using FastQC v0.11.9 (Babraham Bioinformatics, Cambridge) and mapped on the reference genome using Bowtie2 v2.4.2 [69] using default settings. The resulting SAM files were converted to BAM using SAMtools 1.11 [70] and Feature Counts 2.0.1 [71] was used to get the gene counts. The T-REx webserver was used to perform statistical analysis and determine differential gene expressions (DGE) and subsequently, gene set enrichment analysis was done for functional analysis using the GSEA-Pro web server (http://gseapro.molgenrug.nl).

## Biofilm inhibition assay

The bacteria were cultured in CA-MHB medium at 37˚C to an $OD_{600}$ of 1 and then diluted 1:40 into fresh CA-MHB to an $OD_{600}$ of 0.025. 1 ml of bacterial suspension was added into a glass tube and incubated for 8 hours to allow bacteria to adhere to the surface, and then replace the supernatant with a fresh medium containing the indicated concentration of azithromycin or the combination to allow the initial biofilm to grow at 37 ˚C for 16 h. Next, the tube was washed three times with fresh PBS and the biofilm was stained with 1% crystal violet. The crystal violet was dissolved in ethanol and measured at a wavelength of 595 nm. Or the biofilm was dispersed by ultrasound and the bacterial count was determined by plating. All the tests were performed in triplicate.

## Biofilm killing assay

The bacteria were cultured in CA-MHB medium at 37˚C to an $OD_{600}$ of 1 and then diluted 1:40 into fresh CA-MHB to an $OD_{600}$ of 0.025. 1 ml of bacterial suspension was added into a glass tube and incubated at 37˚C for 8 hours to allow bacteria to adhere to the surface, and then replace the supernatant with a fresh medium to allow the initial biofilm to grow at 37 ˚C for 16 h. The tube was washed three times with fresh PBS and added 2 mL fresh CA-MHB with indicated concentration antibiotics and D-11 to cover the biofilms and incubated at 37˚C for 24 h. Next, the tube was washed three times with fresh PBS, the biofilm was dispersed by ultrasound and the bacterial count was determined by plating. All the tests were performed in triplicate.

## Antimicrobial activity in blood

Human blood of healthy individuals was obtained from Sanquin (certified Dutch organization responsible for meeting the need in healthcare for blood and blood products, https://www.sanquin.nl/) and was infected with $5 \times 10^7$ CFU/mL of PAO1. The blood was distributed at 0.5 mL in 2 mL Eppendorf tubes and treated with the antibiotics and peptides at the indicated concentrations at 37 ˚C for 4 hours. The samples were decimal serially diluted in PBS. Drops of 10 μL of each dilution on LB agar to track the development of infection. The experiment was performed in triplicate.

## Cutaneous mouse infection model

The mouse cutaneous abscess infection model was performed as previously described with minor modifications [72,73]. 8-week-old specific-pathogen-free female BALB/c mice were used in the experiment. Before infection, 80 μL of 7.5% chloral hydrate were injected intraperitoneally to anaesthetize the mice. PAO1 was grown to an $OD_{600}$ of 1.0. The bacteria were collected by centrifugation and washed twice with saline. Then the bacteria were resuspended in saline at a concentration of $4 \times 10^8$ CFU/ml. 50 μL of the bacterial suspension were injected

into the right side of the dorsum of the mouse, resulting in $2 \times 10^7$ CFU per mouse. 30 minutes after the infection, 50 μL saline alone or containing indicated concentration of antibiotic or 10 mg/kg D-11, or a combination of antibiotic and 10 mg/kg D-11 was injected into the infected area. 20 hours post infection, the skin abscess was excised and subjected to homogenization in saline. The live bacteria numbers were determined by plating.

## Supporting information

**S1 Fig. The synergistic effect of peptides and macrolides against *P. aeruginosa*.** The FICI plot for synergistic peptides and macrolides against PAO1. AZM, azithromycin; ERY, erythromycin; CLA, clarithromycin; TELI, telithromycin; SPI, spiramycin.
(TIF)

**S2 Fig.** (A) Time killing curves of D-11 against PAO1 during 6 h incubation at the indicated concentration. (B) Survival rates of bacteria under different treatment conditions during PI uptake assay.
(TIF)

**S3 Fig.** Effect of cationic divalent as MgCl$_2$ (**A**) and CaCl$_2$ (**B**) on the synergistic activity of D-11 and azithromycin. The MIC of azithromycin was determined by checkerboard microdilution assays in the presence of MgCl$_2$ or CaCl$_2$ (0–32 mM) with or without 4 μM D-11. Effect of PC (**C**) and PE (**D**) on D-11 and azithromycin combination.
(TIF)

**S4 Fig. Exogenous addition of NAC (6 mM) improved the survival rate of *P. aeruginosa* treated with D-11 and azithromycin.**
(TIF)

**S5 Fig. D-11 enhances the activity of azithromycin in inhibiting biofilm formation.**
(TIF)

**S6 Fig. *Ex vivo* bacteremia model for antibiotics, D-11, and their combinations.** 10 μL of serial decimal dilutions of each one of the combinations of antibiotics with/without D-11 (4 μM) tested in blood were dropped after 4 hours of treatment.
(TIF)

**S7 Fig. Appearance of the treated blood after 24 h.** Black colour of the blood indicates hemolysis and therefore bacterial growth.
(TIF)

**S1 Table. The synergy effects of peptides and macrolides against PAO1.**
(DOCX)

**S2 Table. The results of transcriptome analysis.** Genes that displayed different expression under the treatment of D-11 (**A**), AZM (**B**) or D-11 and AZM combination (**C**) in comparison to the untreatment.
(XLS)

**S3 Table. The screening of synergy effects of D-11 with multiple antibiotics.**
(DOCX)

**S4 Table. The synergy effects of D-11 and antibiotics against different three reference and nine clinical *P. aeruginosa*.**
(XLSX)

**S5 Table. Bacterial strains, plasmids and primers used in this study.**
(DOCX)

**S6 Table. The antibiotic sensibility of the clinical *P. aeruginosa* isolates.**
(DOCX)

## Acknowledgments

The following reagents were obtained through BEI Resources, NIAID, NIH: *P. aeruginosa* strains HM-214, NR-31040 and NR-31041. We thank Prof. Wilbert Bitter in Amsterdam University Medical Center (AUMC) for kindly providing the six clinical *P. aeruginosa* isolates.

## Author Contributions

**Conceptualization:** Yushan Xia, Weihui Wu, Oscar P. Kuipers.

**Data curation:** Yushan Xia, Rubén Cebrián, Congjuan Xu, Anne de Jong.

**Formal analysis:** Yushan Xia, Rubén Cebrián, Congjuan Xu, Anne de Jong.

**Funding acquisition:** Weihui Wu, Oscar P. Kuipers.

**Investigation:** Yushan Xia, Rubén Cebrián, Congjuan Xu, Anne de Jong.

**Methodology:** Yushan Xia, Rubén Cebrián, Weihui Wu, Oscar P. Kuipers.

**Project administration:** Weihui Wu, Oscar P. Kuipers.

**Resources:** Weihui Wu, Oscar P. Kuipers.

**Software:** Yushan Xia, Anne de Jong.

**Supervision:** Weihui Wu, Oscar P. Kuipers.

**Validation:** Yushan Xia.

**Visualization:** Yushan Xia, Rubén Cebrián.

**Writing – original draft:** Yushan Xia, Rubén Cebrián.

**Writing – review & editing:** Weihui Wu, Oscar P. Kuipers.

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
