## [Decision Letter · Decision Letter 0]

30 Jun 2021

Dear Dr. Kuipers,

Thank you very much for submitting your manuscript "Elucidating the mechanism by which synthetic helper peptides sensitize Pseudomonas aeruginosa to multiple antibiotics" for consideration at PLOS Pathogens. As with all papers reviewed by the journal, your manuscript was reviewed by members of the editorial board and by several independent reviewers. In light of the reviews (below this email), we would like to invite the resubmission of a significantly-revised version that takes into account the reviewers' comments.

Please provide a point by point response to all reviewer comments, including the major issues raised by reviewer #1 regarding experimental methodology.

We cannot make any decision about publication until we have seen the revised manuscript and your response to the reviewers' comments. Your revised manuscript is also likely to be sent to reviewers for further evaluation.

Sincerely,

Matthew C Wolfgang

Associate Editor

PLOS Pathogens

Denise Monack

Section Editor

PLOS Pathogens

Kasturi Haldar

Editor-in-Chief

PLOS Pathogens

orcid.org/0000-0001-5065-158X

Michael Malim

Editor-in-Chief

PLOS Pathogens

orcid.org/0000-0002-7699-2064

Please provide a point by point response to all reviewer comments, including the major issues raised by reviewer #1 regarding experimental methodology.

Reviewer's Responses to Questions

**Part I - Summary**

Reviewer #1: The authors describe synergistic effects between peptide D-11 and azithromycin in treatment of p. auruginosa and determine the mechanism of synergy. The strength of the paper is that the authors describe an impressive amount of experiments, and that the potential impact of such studies can be very relevant fro treatment of p. aeruginosa infections.

Unfortunately, several weaknesses are also present. Mostly the methodology used fails to properly indicate that true synergy is happening. Proper controls are lacking to show that the peptide D-11 is specific in any of it's activities.

Especially with respect to antimicrobial activity of the compounds a standard MIC assay and the checkerboard assay as performed is not enough to determine the mechanism of action of peptides and is only a very rough readout of activity and especially of synergy.

Novelty of this study lies mainly in the choice of peptide, but multiple other manuscripts have focussed on the sensitizing effect of (antimicrobial) peptides.

Many questions are

Reviewer #2: The manuscript describes how a synthetic cationic peptide D-11 containing all D-amino acids can exert a synergistic activity when applied with different antibiotics devoid of activity against Gram negative pathogens. Using as model P. aeruginosa, the authors have studied its mechanism of action showing its involvement in the increase of membrane permeability due to its interaction with components of the LPS, the membrane depolarization, reduction of ATP levels, ROS accumulation and impairing efflux pumps, enhancing intracellular accumulation of antibiotics, and proposing a model for the multiple effects that are determined by the interaction of the peptide with both the outer and inner membrane.

Furthermore the authors also extend their study to show the antibiotic effect of these combinations on biofilms, in a ex vivo bacteremia model of infection as well as in a murine model, proposing the potential use of the peptide as adjuvant to potentiate the use of different antibiotics on Gram negative pathogens.

Despite that the combination of peptides with different antibiotics have been already reported to address MDR bacteria, these results are of relevance as they present a detailed study proposing a mechanism of action of the membrane disturbing peptides and the efficacy of the potential application of these combinations in extending the profile of different families of antibiotics currently restricted to Gram positive pathogens, or reducing the MIC of antibiotics with toxicity issues.

Reviewer #3: The topic is timely and will be of interest to the journal's readers. The strength of this work is that the authors provide several lines of evidence to decipher the mechanism of action of D-11 at different levels. The ex vivo and in vivo experiments were done to confirm the efficacy of the synergistic combinations.

The limitations of this work is a very limited number of clinical isolates (PA130, PA179, PA216) and there was no information of drug sensitivity profile of each isolate. Moreover, there was no data on the toxicity of D-11.

**Part II – Major Issues: Key Experiments Required for Acceptance**

Reviewer #1: 1) ) Synergy is based on classic broth dilution assays (including checkerboard) which have a very non-detailed readout after 18h incubation. Most antimicrobial peptides work within 1-2 hours in such assays and then lose activity (from most kinetic figures in this manuscript the effect of D-11 also seems to be mostly within the first hour). Therefore a sub-MIC concentration can (and will) kill a substantial amount of bacteria but the residual number of bacteria (in theory 1 viable bacterium is enough) will grow out to a dense population after 18h. The authors should therefore determine what actually happens in the first hours with respect to bactericidal activity

2) All effects described for D-11 alone , are very similar to many native antimicrobial peptides, including LL-37. This or similar cathelicidins should have been taken along as control. Now I don’t see any indication why D-11 is different from LL-37 (or many other cathelicidins).

3) Transcriptome experiments are done with 8uM Azithromycin. This should have been done at 4 uM since this is the presumed concentration where synergy occured.

4) Biofilm experiments are done very simplistic and lacks characterisation. Specifically:

Specifically:

a) The authors should show and confirm that a proper biofilm is formed (not just 1 or 2 layers of adhered bacteria to glass).

b) Biofilm growth has several stages, it is custom to have an adherence phase of 8 h then replace the supernatant with new media and let the initial biofilm grow out.

c) Crystal violet detection of biofilm mass should be included to get a second readout: biofilm mass (including dead bacteria) and number of viable bacteria in the biofilm.

d) When treatment of a preformed biofilm results in lower number of viable bacteria, are these bacteria killed or just released in the supernatant because the biofilm is broken off. The authors should check this because the latter would actually be a bad characteristic that could increase spreading of biofilms in an in vivo setting.

Reviewer #2: (No Response)

Reviewer #3: There was no major issue.

**Part III – Minor Issues: Editorial and Data Presentation Modifications**

Reviewer #1: Minor:

1) please provide purity of peptides.

2) Fig4C: since this is not an antibiofilm graph I would give this a different spot in the article.

3) Line 180-193: It is not clear (and unlikely) that the interaction with bacterial components is very specific except for the pure electrostatic interaction. If the authors would add another (non bacterial) negatively charged compound a similar effect would be seen I predict. Such controls should be added to the manuscript.

4) Line 170 (and more spots in this paragraph): the figures seem to have been renumbered. Here it says Fig 1D instead of 3D

5) Fig1H in ploymixin should read polymyxin

6) Fig 1H The effect of the control polymyxin seems relatively mild. Another control leading to 100% loss of membrane potential should be included.

7) L592: Bacterial should read bacteria.

8) Line 149: I thought ketolides are a subgroup of macrolides? And why would it be expected that ketolides are not synergistically acting? Or am I misreading this sentence?

9)Line 168: How should PI uptake be interpreted: inner membrane permeabilization (which can synergize with macrolide action), or just killing the bacterium??? In which case it has nothing to do synergy.

10) Fig 4A: Biofilm formation: what is exactly expressed on the y-axis, CFU/OD. What does that mean exactly??

11) Fig 4B If 16 uM Azithromycin already reduces the biofilm by 99% (2LOG) what does a further reduction by D-11 really mean? How relevant is this? Please explain

12) Fig 4D: Activity in blood: it is a nice visual, but why aren’t actual CFU/ml calculated and provided? From the images it seems as if D-11 at 4uM already results in substantial partial killing of P. aeruginosa (compare top line of 0 and 4uM, last three dilutions.

13) the activity in blood clearly shows that sub-MIC D-11 has a clear bactericidal effect . Is the synergy that is measured therefore not just an effect of a lowered number of viable bacteria that have to be killed by the second component (azithromycin)? The MIC of azithromycin at much lower bacterial densities should be determined

14) Different experiments are carried out with very different bacterial densities, ranging from 1x10^6 CFU/ml (MIC assay) to an OD=0.5 corresponding to >10^8 CFU/ml. I understand that higher densities are needed to detect fluorescent signals for membrane potential for example, but the antibacterial effect will be different at these largely different densities and one can not compare and relate the effect of 4 uM D-11 in different assays. Please clarify this lack of consistency and its potential effect on interpretation in the manuscript.

15) The interpretation of the checkerboard assay is unclear to me. Is synergy observed over the whole concentration spectrum of both compounds or (taking D-11 as an example) only at this very specific combination of 4uM Azithromycin and 4uM of D-11?

The FIC is 0.156 based on these partial MICs (4/128 + 4/32), but for example….. how much azithromycin was needed at a D-11 concentration of 2 uM and was that combination synergistic as well. (and that goes for all tested concentrations of course). Please explain.

Reviewer #2: The following minor issues should be addressed to improve the manuscript:

Introduction: Include a description of the peptides that are described in the study (Table 1), and more specially provide any state of the art related to similar studies. Background information should be provided in the introduction regarding these peptides to put the study in context. It is not mentioned where the EC5, KR-12 and L-11 and the rest of peptides in table 1 originate from, references are only provide in the table.

Table 1: it is confusing to read and can be misleading regarding some of the data.

Change order of the columns, including first the MIC of the peptides alone and in combination with azithromycin, then the MIC of azithromycin and in combination with the peptide.

Line 121: Provide clarification regarding the meaning of “reverse sequence of the amino acid”

Line 151: Table 2 is not included in the main text

Line 260: include the concentration used for the combination of D-11 and azithromycin in the transcriptome study. It is mentioned in line 271, but should be included when describing the experiment

Line 264: tables 4-6 are not provided in main text

Line 325 Table 3 is not provided in main text

Efficacy in murine model: no information is provided regarding the concentration and dosage of the combination with different antibiotics in the text. Fig 4 E-G: include the concentration units in all the graphics

Reviewer #3: Please see the reviewer attachment file.

PLOS authors have the option to publish the peer review history of their article (what does this mean?). If published, this will include your full peer review and any attached files.

Reviewer #1: No

Reviewer #2: No

Reviewer #3: No
---

## [Editor Report · Decision Letter 1]

20 Aug 2021

Dear Dr. Kuipers,

We are pleased to inform you that your manuscript 'Elucidating the mechanism by which synthetic helper peptides sensitize Pseudomonas aeruginosa to multiple antibiotics' has been provisionally accepted for publication in PLOS Pathogens.

Best regards,

Matthew C Wolfgang

Associate Editor

PLOS Pathogens

Denise Monack

Section Editor

PLOS Pathogens

Kasturi Haldar

Editor-in-Chief

PLOS Pathogens

orcid.org/0000-0001-5065-158X

Michael Malim

Editor-in-Chief

PLOS Pathogens

orcid.org/0000-0002-7699-2064
---

## [Editor Report · Acceptance letter]

30 Aug 2021

Dear Dr. Kuipers,

We are delighted to inform you that your manuscript, "Elucidating the mechanism by which synthetic helper peptides sensitize *Pseudomonas aeruginosa* to multiple antibiotics," has been formally accepted for publication in PLOS Pathogens.

Best regards,

Kasturi Haldar

Editor-in-Chief

PLOS Pathogens

orcid.org/0000-0001-5065-158X

Michael Malim

Editor-in-Chief

PLOS Pathogens

orcid.org/0000-0002-7699-2064